# Respiratory Tract Cancer Incidences across Industry Groups: A Nationwide Cohort Study with More Than 70 Million Person-Years of Follow-Up

**DOI:** 10.3390/cancers14215219

**Published:** 2022-10-25

**Authors:** Seong-Uk Baek, Woo-Ri Lee, Ki-Bong Yoo, Jun-Hyeok Choi, Kyung-Eun Lee, Wanhyung Lee, Jin-Ha Yoon

**Affiliations:** 1Department of Occupational and Environmental Medicine, Severance Hospital, Yonsei University College of Medicine, Seoul 03722, Korea; 2The Institute for Occupational Health, Yonsei University College of Medicine, Seoul 03722, Korea; 3Graduate School, Yonsei University College of Medicine, Seoul 03722, Korea; 4Division of Cancer Control and Policy, National Cancer Control Institute, National Cancer Center, Goyang 10408, Korea; 5Division of Health Administration, College of Software and Digital Healthcare Convergence, Yonsei University, Wonju 26493, Korea; 6Occupational Safety and Health Research Institute, Korea Occupational Safety and Health Agency, Ulsan 44429, Korea; 7Department of Occupational and Environmental Medicine, Gil Medical Center, Gachon University College of Medicine, Incheon 21565, Korea; 8Graduate School of Public Health, Yonsei University College of Medicine, Seoul 03722, Korea; 9Department of Preventive Medicine, Yonsei University College of Medicine, Seoul 03722, Korea

**Keywords:** laryngeal cancer, lung cancer, carcinogens, industry, standardized incidence ratio

## Abstract

**Simple Summary:**

Our study estimated the risk of laryngeal and lung cancers according to the industrial classification. Certain industries have been found to be vulnerable to respiratory cancer. In particular, workers employed in the land transportation industries have a high risk of laryngeal cancer. Moreover, workers employed in fishing, mining, transportation, construction, animal production, and healthcare industries have a high risk of lung cancer. As an increased risk of respiratory tract cancers has been identified in certain industries in this study, appropriate policy intervention is needed to prevent occupational cancers.

**Abstract:**

The number of cases and incidence rates of laryngeal and lung cancers have been increasing globally. Therefore, identifying the occupational causes of such cancers is an important concern for policymakers to prevent cancers and deaths. We used national health insurance service claims data in Korea. We included 10,786,000 workers aged between 25 and 64 years. In total, 74,366,928 total person-years of follow-up were included in this study with a mean follow-up of 6.89 years for each person. The standardized incidence ratio (SIR) and 95% confidence intervals (CIs) referenced with the total workers were estimated. For laryngeal cancer, increased SIRs were observed in the land transportation industry among male workers (SIR [95% CI]: 1.65 [1.02–2.53]). For lung cancer, elevated SIRs were observed in the industries including animal production (1.72 [1.03–2.68]), fishing (1.70 [1.05–2.60]), mining (1.69 [1.22–2.27]), travel (1.41 [1.00–1.93]), and transportation (1.22 [1.15–1.30]) among male workers. For female works, healthcare (2.08 [1.04–3.72]) and wholesale (1.88 [1.18–2.85]) industries were associated with a high risk of lung cancer. As an increased risk of respiratory tract cancers has been identified in employees associated with certain industries, appropriate policy intervention is needed to prevent occupational cancers.

## 1. Introduction

Analyzing risks and preventing respiratory tract cancers have become an important issue worldwide in recent decades. The number of cases and incidence rates of laryngeal and lung cancers has been increasing globally [1]. Globally, approximately 185,000 and 2.2 million patients were newly diagnosed with laryngeal and lung cancers, respectively, and approximately 100,000 and 1.8 million died from them in 2020 [2]. In South Korea, 1222 and 29,960 patients were newly diagnosed with laryngeal and lung cancers in 2019, accounting for 0.5% and 11.8% of the total cancer cases [3]. In particular, lung cancer, which is a leading cause of cancer deaths, imposes a major burden on both individuals and society [4]. Therefore, identifying the causes of respiratory tract cancers is an important concern for clinical physicians, epidemiologists, and policymakers to prevent cancers and deaths.

So far, certain environmental and occupational factors have been reported to contribute to the development of laryngeal and lung cancers. One British study reported that occupational factors accounted for approximately 2.9% and 21.1% of laryngeal and lung cancers in men, respectively, and 1.6% and 5.3% in women, suggesting that occupational exposure has significant importance in the incidence of respiratory tract cancers [5]. According to the International Agency for Research on Cancer (IARC), several occupational and environmental exposures, including asbestos, heavy metals, outdoor air pollution and particulate matter (PM), silica dust, and welding fumes were identified as lung carcinogens. For laryngeal cancer, asbestos and strong inorganic acid mist were identified as carcinogens with sufficient evidence in humans [6]. Meanwhile, not only hazardous substances, but also some industries themselves were associated with respiratory tract cancers. For example, in studies conducted in USA and northern European countries, the rubber production industry was related to an increased risk of laryngeal and lung cancers among workers [7,8].

Despite the importance of exploring possible carcinogens, uncovering the association between occupational exposure and laryngeal and lung cancers has been limited due to the rarity of such cancers. Our study aimed to estimate the risks of laryngeal and lung cancers for each industrial classification using national public health insurance claim data with more than 70 million person-years of follow-up in Korea [9]. The early diagnosis of laryngeal and lung cancers could promote disease intervention and reduce psychological problems and socioeconomic costs. Therefore, assessing the incidence of respiratory tract cancers in workers exposed to occupational risk factors is an important social issue. Therefore, this study aimed to analyze the differences in the incidence of respiratory tract cancers in certain social domains of industrial groups using big data covering workers in Korea. Our results may contribute to developing appropriate polices for cancer prevention by identifying industries and workers at risk of respiratory tract cancers.

## 2. Materials and Methods

### 2.1. Study Sample

We used the National Health Insurance Service (NHIS) database for analyzing data from 2009 to 2015. The Korean NHIS claims database contains information on all medical use, procedures, and prescriptions of outpatients and inpatients of nearly the entire population of the nation. The NHIS is a mandatory public health insurance service and covers approximately 98% of the nation’s population comprising more than 50 million residents [10]. The NHIS dataset includes individual demographic variables such as age, sex, type of insurance, and industrial category.

In our closed cohort design, we included 10,786,000 workers aged between 25 and 64 years who were observed in 2009, which was the starting point of this study (Table 1). From 2009 to 2015, 74,366,928 total person-years were included in this study with a mean follow-up of 6.89 years for each person.

### 2.2. Cancers

Diseases were defined and coded based on the Korean Standard Classification of Diseases, which corresponds to the 10th revision of International Classification of Diseases (ICD-10). Our target diseases were laryngeal cancer (ICD code: C32) and lung cancer (ICD code: C33, C34). The target cancers were classified for inpatients with claims information with larynx cancer (C32) and lung cancer (C33, C34) as their primary diagnosis.

### 2.3. Industrial Classification

The Korean Standard Industrial Classification (KSIC), which is based on the 4th revision of the International Standard Industrial Classification was used to classify workers’ industrial categories [11]. We first estimated the risk of cancer incidences by industry using the code ‘section’ that classifies the industries into 21 sections (alphabet code), and then using the code ‘group’ that classifies the industries into 232 groups (3-digit code).

### 2.4. Statistical Analysis

For each cancer, the number of cases and follow-up period (person-year) by sex and age groups were calculated. The age-standardized incidence ratio (SIR) and its 95% confidence intervals (CIs) referenced with the total workers were estimated. Age standardization was performed by 5-year unit of age from 25 to 64 years. To calculate SIRs, the number of observed cases was divided by the expected number of cases for each industrial classification, and the CIs were estimated using the Poisson distribution. SIR was calculated for industrial sections or groups with more than 5 cancer cases. Significance was set at *p* < 0.05. All statistical analyses and visualization were performed using the SAS, version 9.4 (SAS Institute, Cary, NC, USA) and R software (version 4.2.1; R Foundation for Statistical Computing, Vienna, Austria).

### 2.5. Ethics Statement

The NHIS dataset was anonymized before the release of the NHIS dataset to the authors. The Institutional Review Board of the Yonsei Health System reviewed and approved the study (IRB number: Y-2017-0100).

## 3. Results

Out of the total 10,786,000 included workers, 1150 and 15,013 workers were newly diagnosed with laryngeal and lung cancers, respectively. Laryngeal cancer was recorded in 1123 male workers and 27 female workers. Meanwhile, lung cancer was recorded in 12,270 male workers and 2743 female workers. In both laryngeal and lung cancers, the incidence rate increased as the age of workers increased (Table 2).

Figure 1 presents the SIRs of laryngeal cancer compared to the whole workers in each industrial section. Industrial section ‘E. Water supply; sewage, waste management, materials recovery’ and ‘U. Activities of extraterritorial organizations and bodies’ were omitted due to the lack of incidence in these sections. Sex stratification was performed only for male workers because of the low incidence of laryngeal cancer among female workers. Only ‘H. Transportation and storage’ section (SIR [95% CI]: 1.19 [1.00–1.42]) was marginally associated with laryngeal cancer.

Figure 2 depicts the SIRs of lung cancer compared to all workers in each industrial section. Among male workers, significantly increased SIRs were observed in the ‘A. Agriculture, forestry, and fishing’ (SIR [95% CI]: 1.50 [1.17–1.90]), ‘B. Mining and quarrying’ (SIR [95% CI]: 1.44 [1.11–1.85]), ‘F. Construction’ (SIR [95% CI]: 1.07 [1.01–1.13]), ‘H. Transportation and storage’ (SIR [95% CI]: 1.17 [1.11–1.24]), ‘N. Business facilities management and business support services; rental and leasing activities’ (SIR [95% CI]: 1.09 [1.02–1.17]), and ‘O. Public administration and defense; compulsory social security’ sections (SIR [95% CI]: 1.21 [1.00–1.45]). Among female workers, the ‘G. Wholesale and retail trade’ (SIR [95% CI]: 1.13 [1.00–1.26]) section was associated with a high risk of lung cancer.

Table 3 presents the industrial group with a significantly higher risk of laryngeal cancer among male workers. Significantly increased SIR was observed in the ‘passenger land transport, except transport via railways’ (SIR [95% CI]: 1.65 [1.02–2.53]) group. SIRs of laryngeal cancer for other industries are shown in Appendix A.

Table 4 presents the industrial groups with a significantly higher risk of lung cancer. Among male workers, significantly increased SIRs were observed in the ‘animal production’ (SIR [95% CI]: 1.72 [1.03–2.68]), ‘fishing’ (SIR [95% CI]: 1.70 [1.05–2.60]), ‘mining of coal and lignite’ (SIR [95% CI]: 1.69 [1.22–2.27]), ‘activities of travel agencies and tour operators and tourist assistance activities’ (SIR [95% CI]: 1.41 [1.00–1.93]), ‘cleaning and pest control services of building and industrial facilities’ (SIR [95% CI]: 1.23 [1.04–1.46]), ‘support activities for transportation’ (SIR [95% CI]: 1.22 [1.15–1.30]), ‘Heavy and civil engineering construction’ (SIR [95% CI]: 1.17 [1.05–1.31]), and ‘real estate activities on a fee or contract basis’ (SIR [95% CI]: 1.11 [1.01–1.23]). 

Among female workers, significantly increased SIRs were observed in the ‘other human health activities’ (SIR [95% CI]: 2.08 [1.04–3.72]), ‘wholesale of machinery, equipment, and supplies’ (SIR [95% CI]: 1.88 [1.18–2.85]), and ‘hospital activities’ (SIR [95% CI]: 1.29 [1.01–1.63]). SIRs of lung cancer for other industries are shown in Appendix A.

## 4. Discussion

This study aimed to investigate the risks of laryngeal and lung cancers for each industrial classification. The results revealed a diverse cancer incidence across industries. When referenced with total workers, employees in the land transportation industry were associated with a risk of laryngeal cancer. In the case of lung cancer, workers in animal production, fishing, mining, travel activities, transportation, construction, real estate activities, wholesale, and healthcare industries were associated with lung cancer. 

Overall, both laryngeal and lung cancer incidence rates increased with age. This result is in line with previous epidemiological studies [3,12]. The high incidence of cancer in the elderly can be attributable to the effect of chronic conditions or accumulated years of risky health behaviors such as smoking and alcohol drinking [13]. In addition, workers were more likely to be exposed to occupational carcinogens such as asbestos in past working conditions [14].

### 4.1. Laryngeal Cancer

Previous research has revealed an increased risk of laryngeal cancer among workers in several industrial sectors, including the automobile, railroad, lumber, food, and rubber industries [7,15,16]. Additionally, blue-collar workers (production workers, transport equipment operators, and laborers) were reportedly at a higher risk of laryngeal cancer, while white-collar workers were reportedly at a lower risk of laryngeal cancer according to a previous meta-analysis [17]. Overall, our results are consistent with previous findings that workers in the transportation industries have an increased risk of laryngeal cancer [18].

For workers in the transportation industry, occupational exposure to engine exhaust and ambient air pollution might contribute to an increased risk of laryngeal cancer. Previous studies investigated whether occupational exposure to diesel and gasoline exhaust can induce laryngeal cancer. For instance, a Canadian study has identified an elevated relative risk (RR) of laryngeal cancer (RR [95% CI]: 1.59 [1.08–2.33]) [19]. According to a Turkish case–control study, increased odds ratios (ORs) of laryngeal cancer was observed for occupational exposure to both diesel exhaust (OR [95% CI]: 1.5 [1.3–1.9]) and gasoline exhaust (OR [95% CI]: 1.6 [1.3–2.0] [20]. A previous study that estimated meta-RR revealed that the presence of engine exhaust is significantly associated with laryngeal cancer [21]. Additionally, few studies have suggested an association between air pollutants and PM and laryngeal cancer. Air pollutants such as PM_10_, PM_2.5_, and NO_2_ are related to increased risks of laryngeal cancer [22,23,24]. In line with prior findings, we revealed that workers in the land transport industry (e.g., bus drivers and taxi drivers) are at risk of laryngeal cancer.

### 4.2. Lung Cancer

Previous research has revealed that workers in various industries including the rubber manufacturing, petroleum, iron/steel foundry, and nuclear industries were at risk of lung cancer [8,25,26]. In our study, an increased risk of lung cancer was also associated with employees working in a wider variety of industries compared with laryngeal cancer.

Industries that may be exposed to occupational dust such as crystalline silica and asbestos include the mining and construction industries. Thus, our findings support the results of previous studies that reported an association between these industries and lung cancer [27,28]. In addition to occupational dust, these workers are highly likely to be exposed to carcinogens such as radon gas, polycyclic aromatic hydrocarbons (PAHs), and welding fumes.

In the case of the transportation industry, significantly increased SIR was observed for lung cancer. Our results are consistent with the results of recent previous studies that revealed a significant relationship between transportation and lung cancer risk [29,30]. Respiratory carcinogens, such as diesel/gasoline engine exhaust, air pollution, and PM, have been reported as factors that may expose workers in the transportation industry. The increased risk of lung cancer among travel assistance activities could be also attributable to characteristics similar to those in the transportation industry. Workers in the travel agency industry often participate in providing transportation services for tourists [31], and because of this similarity, this industry was included in the transportation industry before the recent KSIC revision.

Several recent studies have reported the relationship between the fishing industry and lung cancer. For instance, a Norwegian epidemiological study observed a high risk of lung cancer among seafarers and fishermen [32], and a Japanese study reported a high risk of lung cancer among workers in the fishing industry [33]. Carcinogens such as diesel engine exhaust, PAHs, and benzene that can be present in ships have also been considered the cause of the high lung cancer incidence among workers in the fishing industry [34].

In our study, the high lung cancer risk was observed among male workers employed in the animal production industry group, which contradicts previous reports that state that farm animal exposure reduces the risk of lung cancer [35]. Moreover, it should be considered that confounding variables such as smoking were not adjusted in this study. Nevertheless, a study has reported that occupational exposure to animal contact or organic dust of animal origin increases the risk of lung cancer [36]. Moreover, our findings were in line with previous studies that showed that workers in the agriculture industry were related to a high risk of lung cancer [37,38]. Occupational exposure to pesticides has been pointed out as the main causes of high lung cancer incidences among agriculture workers [37,39].

A previous case–control study revealed that professional cleaning activities were associated with high risk of lung cancer among female workers [40]. Workers in the ‘cleaning and pest control services’ industry could be exposed to pesticides and cleaning agents, which were carcinogen to lung cancer [41,42]. Workers in the real estate activities industry were reported to have a high risk of lung cancer mortality in preceding studies [30,43]. On the other hand, another study reported a protective effect of this industry on lung cancer [44]. It is unclear whether workers in the real estate activities industry are exposed to occupational carcinogens, so caution is needed in interpretation of these results.

In the case of lung cancer among female workers, our findings support previous studies that revealed a high risk of lung cancer among workers in the healthcare industry [43,45]. Occupational exposure to ionizing radiation or night shifts might be possible explanations [46,47]. Moreover, our findings support that of a previous study that revealed an increased risk in lung cancer mortality among female workers in the wholesale sector, possibly due to the exposure of dust containing silica from the machines or diesel exhausts in the storage warehouse [43].

An insignificantly increased risk was observed in workers in the petroleum, steel manufacturing, and metal foundry industries, which had been reported to be related to lung cancer in previous studies. These results should be considered along with the relatively short follow-up period of this study and the changes in the working environment over the decades.

### 4.3. Limitations

Although our study was based on a large sample size from the NHIS dataset, the findings should be interpreted with caution due to the following limitations. First, we did not adjust for important confounding factors such as smoking and drinking due to lack of information. Nevertheless, a recent study by Jung et al. that analyzed the smoking status of Korean workers provides a glimpse into the confounding effect of smoking on the association between respiratory tract cancer and industrial groups [48]. According to the study, in male workers, the prevalence of smoking was higher among the workers in the fishing, mining, and construction industries than that of overall male workers. On the other hand, the smoking prevalence of the workers in the industries of animal production, travel agencies, cleaning and pest control services, and real estate activities did not differ notably from that of overall workers. Moreover, in the case of women, a lower prevalence of smoking was observed in healthcare workers compared to overall female workers. Furthermore, alcohol beverages are classified as a carcinogenic agent with sufficient evidence in humans (IARC group 1) for laryngeal cancer [49]. Therefore, drinking habit can act as a strong confounding factor, especially for laryngeal cancer. In addition, environmental factors including residential areas, outdoor particulate matter exposures, and second-hand tobacco smoking, and individual respiratory disease history can have a confounding effect on the relationship between respiratory tract cancer incidences and industrial groups. Nevertheless, the purpose of the current study was to identify vulnerable industries that may need intervention and to provide information that can help policymakers implement occupational safety and health policies. Further in-depth studies are needed to confirm whether the risk of laryngeal and lung cancers is independent of the confounding factors. Second, we could not consider which industry the workers were engaged in before 2009. Additionally, the trend of the risk according to the period of employment in the industry could not be estimated. Third, our study only presented risks according to the industrial classification, and specific occupational carcinogens could not be specified. Further studies with exposure data are needed, and an alternative job exposure matrix using industrial classification should be used to clarify the association between cancer incidence and industrial classification. Fourth, it should also be considered that, even in the same industry, the risk of cancer incidence changes with the times due to changes in working environments [8].

## 5. Conclusions

Our study estimated the risk of laryngeal and lung cancers according to the industrial classification and suggested the at-risk industries. Certain industries have been revealed as vulnerable social domains for respiratory tract cancers. In particular, workers employed in the transportation have a high risk of both laryngeal and lung cancers. Despite the recent developments of occupational environments, an increased risk of respiratory tract cancers has been identified in certain industries in this study. Therefore, appropriate policy intervention is needed to prevent occupational cancers.

## Figures and Tables

**Figure 1 cancers-14-05219-f001:**
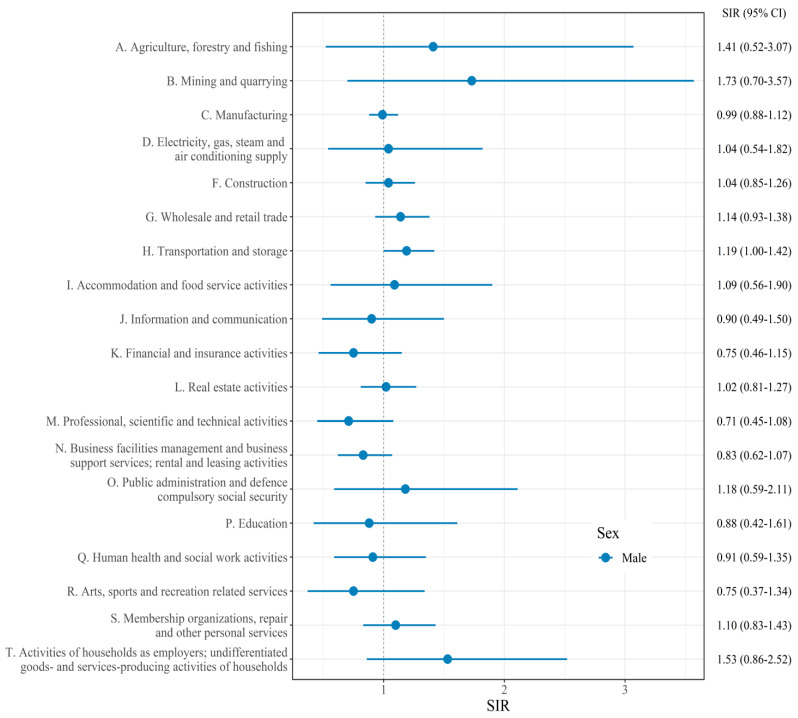
Standardized incidence ratio (SIR) and confidence intervals (CIs) of laryngeal cancer for each industry section.

**Figure 2 cancers-14-05219-f002:**
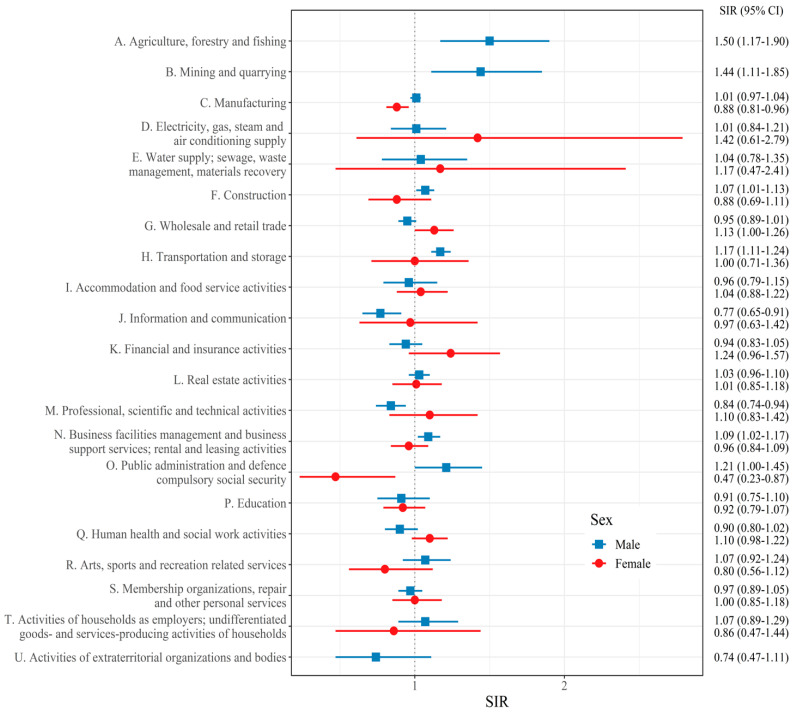
Standardized incidence ratio (SIR) and confidence intervals (CIs) of lung cancer for each industry section.

**Table 1 cancers-14-05219-t001:** Baseline distribution of age group and included number of workers.

	Male		Female	
	N	%	N	%
Total workers	7,167,927	100	3,618,073	100
Age				
25–29	900,930	12.6	871,625	24.1
30–34	1,201,461	16.8	652,619	18.0
35–39	1,326,709	18.5	563,564	15.6
40–44	1,156,361	16.1	513,724	14.2
45–49	1,022,191	14.3	444,480	12.3
50–54	810,748	11.3	315,289	8.7
55–59	474,025	6.6	169,750	4.7
60–64	275,502	3.8	87,022	2.4
Year				
2009	7,167,927	100	3,618,073	100
2010	7,116,267	99.3	3,601,730	99.5
2011	7,076,872	98.7	3,590,469	99.2
2012	7,024,114	98.0	3,580,963	99.0
2013	6,970,061	97.2	3,571,713	98.7
2014	6,943,079	96.9	3,568,022	98.6
2015	6,925,839	96.6	3,565,209	98.5

**Table 2 cancers-14-05219-t002:** Follow-up period and number of incidence cancer cases according to sex and age group.

		Larynx Cancer (C32)			Lung Cancer (C33, C34)		
		Person-Year	Cases	Incidence Rate (per 100,000)	Person-Year	Cases	Incidence Rate (per 100,000)
Total		74,366,928	1150	1.54	71,872,786	15,013	20.08
Sex	Male	49,253,195	1123	2.27	49,231,565	12,270	24.74
	Female	25,113,734	27	0.11	25,107,016	2743	10.09
Age	25–29	12,143,876	7	0.06	12,143,561	149	1.20
	30–34	12,766,801	9	0.07	12,766,145	315	2.43
	35–39	13,035,008	37	0.28	13,033,646	656	4.98
	40–44	11,568,741	60	0.52	11,566,159	1232	10.55
	45–49	10,170,955	147	1.47	10,166,540	2172	21.13
	50–54	7,783,530	274	3.52	7,777,179	3297	41.94
	55–59	4,426,098	324	7.28	4,419,556	3694	82.66
	60–64	2,471,919	292	11.77	2,465,795	3626	145.98

**Table 3 cancers-14-05219-t003:** Industry groups with a significantly higher risk of laryngeal cancer.

KSIC Code	Industrial Classification (Group)	Cases	Person-Year	SIR (95% CI)
**Male Workers**				
492	Passenger land transport, except transport via railways	21	308,510	1.65 (1.02–2.53)

KSIC, Korean Standard Industrial Classification; SIR, standardized incidence ratio; CI, confidence interval.

**Table 4 cancers-14-05219-t004:** Industry groups with a significantly higher risk of lung cancer.

KSIC Code	Industrial Classification (Group)	Cases	Person-Year	SIR (95% CI)
**Male Workers**				
012	Animal production	19	36,266	1.72 (1.03–2.68)
031	Fishing	21	33,588	1.70 (1.05–2.60)
051	Mining of coal and lignite	43	68,719	1.69 (1.22–2.27)
752	Activities of travel agencies and tour operators and tourist assistance activities	39	75,153	1.41 (1.00–1.93)
742	Cleaning and pest control services of building and industrial facilities	139	169,980	1.23 (1.04–1.46)
529	Support activities for transportation	1061	2,183,034	1.22 (1.15–1.30)
412	Heavy and civil engineering construction	308	966,047	1.17 (1.05–1.31)
682	Real estate activities on a fee or contract basis	407	563,482	1.11 (1.01–1.23)
**Female workers**				
869	Other human health activities	11	41,399	2.08 (1.04–3.72)
465	Wholesale of machinery, equipment, and supplies	22	118,943	1.88 (1.18–2.85)
861	Hospital activities	70	637,586	1.29 (1.01–1.63)

KSIC, Korean Standard Industrial Classification; SIR, standardized incidence ratio; CI, confidence interval.

## Data Availability

The data are not publicly available.

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
