# Peer review of "Respiratory Tract Cancer Incidences across Industry Groups: A Nationwide Cohort Study with More Than 70 Million Person-Years of Follow-Up"

_cancers, 2022, doi:10.3390/cancers14215219_

Round 1
Reviewer 1 Report
Respiratory tract cancer incidences across industry groups: a nationwide cohort study with more than 70 million person-years 3 of follow-up
This study of respiratory tract cancer incidence across industry groups, is a large cohort study that looks at diverse occupations and does a good job of delineating the impact of industry group on cancer risk.
Introduction
Line 49-51; are the 2020 numbers global or for Korea? Please specify. I would recommend including the current status for number of cases in Korea specifically here as well.
Line 67: Specify study population for rubber production laryngeal and lung cancer findings.
Discussion:
The effect of age on cancer onset/diagnoses is missing in the discussion. In the results, you note that SIR increase with increasing age.
The importance of pesticides/insecticides/ other agricultural chemicals that have been shown to increase cancer risk in agricultural workers should be considered. You do address cleaners and pest control but not agriculture.
Main limitation: this study does not adjust for smoking. The lack of smoking data makes it very hard to delineate the true impact of industry group in this analysis. a limitation you address in the manuscript. I think adding a bit more information about smoking/drinking trends in the occupational groups by age groups where you had significant findings might be useful.
Author Response
Dear editor and reviewers,
Authors thank to the editor and reviewers for their valuable feedback on our submission to Cancers and giving us the opportunity to submit a revised manuscript “Respiratory tract cancer incidences across industry groups: a nationwide cohort study with more than 70 million person-years of follow-up”. Please see our point-by-point response to each concern. Thank you.
Reviewers' Comments to the Authors:
Reviewer #1:
Reviewer #1: This study of respiratory tract cancer incidence across industry groups, is a large cohort study that looks at diverse occupations and does a good job of delineating the impact of industry group on cancer risk.
Response) Thank you for your constructive suggestions for our study. Your insightful comments were very helpful in reshaping the manuscript. Please see our point-by-point response to each concern. Thank you.
Introduction
Point 1 Line 49-51; are the 2020 numbers global or for Korea? Please specify. I would recommend including the current status for number of cases in Korea specifically here as well.
Response 1) Thank you. We clarifed the meaning of the sentece. Further, we included the current status for the number of cancer cases in Korea.
Line 49-53; Globally, approximately 185,000 and 2.2 million patients were newly diagnosed with laryngeal and lung cancers, respectively, and approximately 100,000 and 1.8 million died from them in 2020 [1]. In South Korea, 1222 and 29,960 patients were newly diagnosed with laryngeal and lung cancers in 2019, accounting for 0.5% and 11.8% of the total cancer cases [2].
Point 2 Line 67: Specify study population for rubber production laryngeal and lung cancer findings.
Response 2) Thank you for the suggestion. We specified the study population for the rubber production laryngeal and lung cancer findings.
Line 68-71; For example, in studies conducted in USA and Northern European countries, the rubber production industry was related to an increased risk of laryngeal and lung cancers among workers
Discussion
Point 3 The effect of age on cancer onset/diagnoses is missing in the discussion. In the results, you note that SIR increase with increasing age.
Response 3) Thank you for the suggestion. We mentioned the effect of age on cancer onset in Discussion section.
Line 183-188; Overall, both laryngeal and lung cancer incidence rates increased with age. This result is in line with previous epidemiological studies [2, 3]. The high incidence of cancer in the elderly can be attributable to the effect of chronic conditions or accumulated years of risky health behaviors such as smoking and alcohol drinking [4]. In addition, workers were more likely to be exposed to occupational carcinogens such as asbestos in past working conditions [5].
Point 4 The importance of pesticides/insecticides/ other agricultural chemicals that have been shown to increase cancer risk in agricultural workers should be considered. You do address cleaners and pest control but not agriculture.
Response 4) Thank you for the suggestion. We addressed the recent researches on lung cancer risk of agriculture worekers and the effect of pesticides.
Line 244-248; Also, our findings were in line with previous studies that showed that workers in agriculture industry were related to a high risk of lung cancer [6, 7]. Occupational exposure to pesticides has been pointed out as the main causes of high lung cancer incidences among agriculture workers [6, 8].
Point 5 Main limitation: this study does not adjust for smoking. The lack of smoking data makes it very hard to delineate the true impact of industry group in this analysis. a limitation you address in the manuscript. I think adding a bit more information about smoking/drinking trends in the occupational groups by age groups where you had significant findings might be useful.
Response 5) Thank you for your valuable suggestion. In the reviesed manuscript, we cited a relevent study that examined the smoking status of Korean workers by each industrial group. In doing so, we further described how the association between cancer incidences and industry groups observed in this study could be affected by smoking. We were able to interpret in more depth with the relevent study on the smoking status of Korean workers. We believe this description will give meaningful information to readers to interpret our findings.
Line 273-285; Nevertheless, a recent study by Jung et al. that analyzed the smoking status of Korean workers provides a glimpse into the confounding effect of smoking on the association between respiratory tract cancer and industiral groups [9]. According to the study, in male workers, the prevalence of smoking was higher in the workers in fishing, mining, and construction industries than that of overall male workers. On the other hand, the smoking prevalence of the workers in the industries of animal production, travel agencies, cleaning and pest control services, and real estate activities did not differ notably from that of overall workers. Also, in the case of women, a lower prevalence of smoking was observed in healthcare workers compared to overall female workers. The purpose of the current study was identifying vulnerable industries that may need intervention and and providing information that can help policymakers implement occupational safety and health policies. Further in-depth studies are needed to confirm whether the risk of laryngeal and lung cancers is independent of the confounding factors.
Reviewer 2 Report
The paper entitled:"Respiratory tract cancer incidences across industry groups: a nationwide cohort study with more than 70 million person-years 3 of follow-up" presents a several important findings for a nationwide sample. The paper is well written and its structure is easy to follow. The authors have done an interested work, although further analysis of confounding factors such as smoking and alcohol intake would have give more scientific soundness to this paper.
Author Response
Dear editor and reviewers,
Authors thank to the editor and reviewers for their valuable feedback on our submission to Cancers and giving us the opportunity to submit a revised manuscript “Respiratory tract cancer incidences across industry groups: a nationwide cohort study with more than 70 million person-years of follow-up”. Please see our point-by-point response to each concern. Thank you.
Reviewer #2:
The paper entitled:"Respiratory tract cancer incidences across industry groups: a nationwide cohort study with more than 70 million person-years 3 of follow-up" presents a several important findings for a nationwide sample. The paper is well written and its structure is easy to follow. The authors have done an interested work, although further analysis of confounding factors such as smoking and alcohol intake would have give more scientific soundness to this paper.
Response) Thank you for your valuable suggestion. In the reviesed manuscript, we cited a relevent study that examined the smoking status of Korean workers by each industrial group. In doing so, we further described how the association between cancer incidences and industry groups observed in this study could be affected by smoking. We were able to interpret in more depth with the relevent study on the smoking status of Korean workers. We believe this description will give meaningful information to readers to interpret our findings.
Line 273-285; Nevertheless, a recent study by Jung et al. that analyzed the smoking status of Korean workers provides a glimpse into the confounding effect of smoking on the association between respiratory tract cancer and industiral groups [1]. According to the study, in male workers, the prevalence of smoking was higher in the workers in fishing, mining, and construction industries than that of overall male workers. On the other hand, the smoking prevalence of the workers in the industries of animal production, travel agencies, cleaning and pest control services, and real estate activities did not differ notably from that of overall workers. Also, in the case of women, a lower prevalence of smoking was observed in healthcare workers compared to overall female workers. The purpose of the current study was identifying vulnerable industries that may need intervention and and providing information that can help policymakers implement occupational safety and health policies. Further in-depth studies are needed to confirm whether the risk of laryngeal and lung cancers is independent of the confounding factors.

Reviewer 3 Report
Authors used insurance service database to determine incidences of the respiratory tract cancers across different industry group between 2009-2015.
Results presented in the manuscript might help guide policy maker for better policy intervention to create better occuptaional environment and inform workers about the hazards associated with working industry.
Biologically, results presented in this manuscripts are not conclusive but mere correlation and suffers with dozens of confounding factors as mentioned in the limitations section of the manuscript and many more.
The study itself does not have any flaws but the findings are not impactful.
Author Response
Dear editor and reviewers,
Authors thank to the editor and reviewers for their valuable feedback on our submission to Cancers and giving us the opportunity to submit a revised manuscript “Respiratory tract cancer incidences across industry groups: a nationwide cohort study with more than 70 million person-years of follow-up”. Please see our point-by-point response to each concern. Thank you.
Reviewer #3:
Authors used insurance service database to determine incidences of the respiratory tract cancers across different industry group between 2009-2015.
Results presented in the manuscript might help guide policy maker for better policy intervention to create better occuptaional environment and inform workers about the hazards associated with working industry.
Biologically, results presented in this manuscripts are not conclusive but mere correlation and suffers with dozens of confounding factors as mentioned in the limitations section of the manuscript and many more.
The study itself does not have any flaws but the findings are not impactful.
Response) Thank you for your valuable suggestion. In the revised manuscript, we cited a relevant study that examined the smoking status of Korean workers by each industrial group. In doing so, we further described how the association between cancer incidences and industry groups observed in this study could be affected by smoking. We were able to interpret in more depth with the relevant study on the smoking status of Korean workers. We believe this description will give meaningful information to readers to interpret our findings. Also, please consider that our study aim was to provide basic information on the association between respiratory tract cancer incidences and diverse industries. We believe that many following in-depth researches will focus on specific industries based on our findings.
Line 273-285; Nevertheless, a recent study by Jung et al. that analyzed the smoking status of Korean workers provides a glimpse into the confounding effect of smoking on the association between respiratory tract cancer and industrial groups [1]. According to the study, in male workers, the prevalence of smoking was higher in the workers in fishing, mining, and construction industries than that of overall male workers. On the other hand, the smoking prevalence of the workers in the industries of animal production, travel agencies, cleaning and pest control services, and real estate activities did not differ notably from that of overall workers. Also, in the case of women, a lower prevalence of smoking was observed in healthcare workers compared to overall female workers. The purpose of the current study was identifying vulnerable industries that may need intervention and and providing information that can help policymakers implement occupational safety and health policies. Further in-depth studies are needed to confirm whether the risk of laryngeal and lung cancers is independent of the confounding factors.

Round 2
Reviewer 3 Report
The revised version of manuscript has incorporated the limitations of the study and has stated some confounding factors.
It is required to address the confounding factors so that readers can draw correct interpretations of the findings. Therefore, it is recommended that authors expand a little more on limitation section and briefly mention all possible confounding factors in their study for each cancer type.
Author Response
Dear editor and reviewers,
Authors thank to the editor and reviewers for their valuable feedback on our submission to Cancers and giving us the opportunity to submit a revised manuscript “Respiratory tract cancer incidences across industry groups: a nationwide cohort study with more than 70 million person-years of follow-up”. Please see our response to your concern. Modifications were highlighted in the revised manuscirpt. Thank you.
- Reviewer #3
The revised version of manuscript has incorporated the limitations of the study and has stated some confounding factors.
It is required to address the confounding factors so that readers can draw correct interpretations of the findings. Therefore, it is recommended that authors expand a little more on limitation section and briefly mention all possible confounding factors in their study for each cancer type.
Authors' response) Thank you for your valuable suggestion. In our revised manuscript, we additionally mentioned some possible confounders that can affect our findings. We expanded a little more on the limitation section.
Line 281-287: Also, alcohol beverages are classified as a carcinogenic agent with sufficient evidence in humans (IARC group 1) for laryngeal cancer [49]. Therefore, drinking habit can act as a strong confounding factor, especially for laryngeal cancer. In addition, environmental factors including residential areas, outdoor particulate matter exposures, and seconhand tobacco smoking; and individual respiratory disease history can have a confounding effect on the relationship between respiratory tract cancer incidences and industrial groups.
Thank you.